# PASA: Proteomic analysis of serum antibodies web server

**Oren Avram**[id][⊙], **Aya Kigel**[⊙], **Anna Vaisman-Mentesh**[id], **Sharon Kligsberg**[id], **Shai Rosenstein, Yael Dror, Tal Pupko**[id], **Yariv Wine**[id] *

The Shmunis School of Biomedicine and Cancer Research, George S. Wise Faculty of Life Sciences, Tel Aviv University, Tel Aviv, Israel

⊙ These authors contributed equally to this work.
* YarivWine@tauex.tau.ac.il

**Data Availability Statement:** Availability and implementation: PASA is freely available for noncommercial users as a web server at https://pasa.tau.ac.il. The source code of this project is

## Abstract

### Motivation

A comprehensive characterization of the humoral response towards a specific antigen requires quantification of the B-cell receptor repertoire by next-generation sequencing (BCR-Seq), as well as the analysis of serum antibodies against this antigen, using proteomics. The proteomic analysis is challenging since it necessitates the mapping of antigen-specific peptides to individual B-cell clones.

### Results

The PASA web server provides a robust computational platform for the analysis and integration of data obtained from proteomics of serum antibodies. PASA maps peptides derived from antibodies raised against a specific antigen to corresponding antibody sequences. It then analyzes and integrates proteomics and BCR-Seq data, thus providing a comprehensive characterization of the humoral response. The PASA web server is freely available at https://pasa.tau.ac.il and open to all users without a login requirement.

## Author summary

The proteomics of serum antibodies (Ig-Seq) is based on a two-arm foundation: the generation of BCR-Seq (using next-generation sequencing of B cells) and high-resolution mass-spectrometry (LC-MS/MS) of serum antibodies. The BCR-Seq data are used for the interpretation of mass-spectra and result in the identification of antibody derived peptides thus, the identification of V genes of the serum antibodies. To provide accessibility to non-expert users we introduce the PASA (Proteomic Analysis of Serum Antibodies) web server that provides a robust computational platform for the analysis and integration of data obtained from proteomics of serum antibodies and enables its integration with BCR-Seq data as obtained from ASAP (that provides bioinformatics support for BCR-Seq analysis and also provides as output a reference database that can be used in proteomic analysis of serum antibodies. ASAP is available at http://asap.tau.ac.il. PASA has a

written in Python and is available at https://github.com/orenavram/PASA.

**Funding:** Y.W. funded by Israel Science Foundation (ISF) #1282/17 Yariv Wine, isf.org.il O.A. was supported in part by the Dalia and Eli Hurvits foundation and by a fellowship from the Edmond J. Safra Center for Bioinformatics at Tel Aviv University. The funders had no role in study design, data collection and analysis, decision to publish, or preparation of the manuscript.

**Competing interests:** The authors have declared that no competing interests exist.

user-friendly interface, while it keeps the ability of expert users to tune their computations towards their special needs. We provide a GALLERY section that demonstrates selected outputs of PASA and example data to allow a quick ramp up for new users. Our graphical outputs promote quantitative analyses and visualization of the immune repertoires.

This is a *PLOS Computational Biology* Software paper.

## Introduction

The hallmark of the adaptive immune response is based on its ability to generate an enormous diversity of different monoclonal antibodies mainly by chromosomal rearrangement, somatic hypermutation (SHM), and class-switch recombination. Antibody diversity occurs within the variable regions of the heavy and light chains ($V_H$ and $V_L$, respectively), with the highest diversity in the complementary determining region 3 of the heavy chain (CDRH3) [1,2]. CDRH3 is often used as a unique identifier to determine antibody clonality [3,4]. Next-generation sequencing (NGS) of B cell receptors (BCR-Seq) revolutionized our ability to capture the diversity of antibodies at the highest resolution and helped address important immunological questions related to the development of the adaptive immune response in health and disease [4,5]. We and others have previously developed computational tools for analyzing the massive NGS data [6,7] that are *continuously being* accumulated in the The Adaptive Immune Receptor Repertoire (AIRR) distributed repository for public use [8]. However, BCR-Seq alone does not reveal the identity, relative concentration, and clonality of antigen-specific antibodies in the blood and/or secretions. Such data are highly important in order to address fundamental immunological questions and to institute a comprehensive immunome map [9]. This knowledge gap can be approached by using high-resolution shotgun tandem mass-spectrometry (LC-MS/MS, Ig-Seq) [2,10,11]. Since its establishment [2], Ig-Seq has opened new research avenues that yielded invaluable insights regarding the development of the immune response during disease and following vaccination [12–17]. The proteomic deconvolution of serum antibodies is challenging due to the extensive antibody diversification resulting from chromosomal rearrangements and SHM, which makes the mapping of obtained antibody-derived peptides to antibody germline sequences irrelevant. Therefore, obtaining the reference BCR-Seq database for each individual is a prerequisite for analyzing LC-MS/MS antibody datasets. Here, we developed PASA, an integrative approach that combines NGS with serum-antibody proteomics (derived from either human or murine), which allows mapping, analysis, and integration of peptides data with reference to individual-specific BCR-Seq data.

## Design and *implementation*

The PASA web server is schematically illustrated in Fig 1. The input of PASA is: (1) BCR-Seq data, as obtained from ASAP [6]; (2) The raw mass spectrometry data files as obtained from LC-MS/MS [11,15]; (3) As part of the advanced options, the user can specify the digestion enzyme used to proteolytically cleave the antibodies (by default, Trypsin). Instead of (2), the user can provide a file of derived peptides obtained using MaxQuant [18,19]. As part of the proteomics experiments, antigen-specific antibodies are enriched by affinity chromatography against a given antigen. The enriched fraction is called "elution" and the depleted fraction is

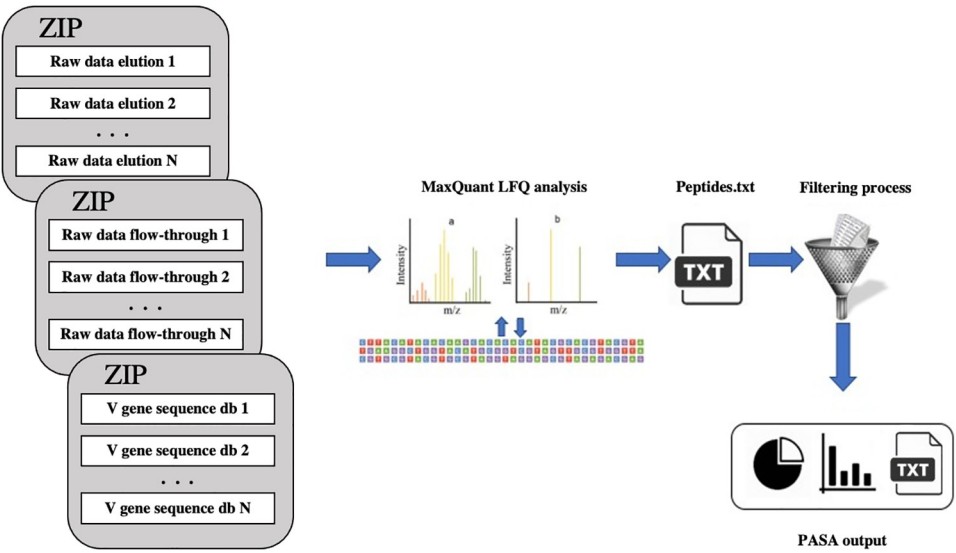

**Fig 1. PASA web server.** PASA accepts (two or more) zipped proteomics spectral files in raw format. For the integration of data derived from NGS with data of antibody encoding B cells, the user is requested to upload V gene sequence database at the amino acid level. Our methodology is divided into two parts. The first part utilizes MaxQuant [18,19] to identify the sequences of peptides derived from antigen-specific serum antibodies. In the second part, mapped serum antibodies are analyzed to provide repertoire measurements and integrated with BCR-Seq data to establish a comprehensive immune map of the antibody response.

called "flow-through". Peptides that are enriched in the elution fraction compared to the flow-through are considered antigen-specific. In order to use label-free quantification (LFQ) as implemented in MaxQuant, PASA requires at least two replicates for each affinity chromatography fraction. Example input files are provided in the web server. PASA's running time may vary as it depends on the number of replicates and databases used. For instance, the analysis of the example input files (elution and flow-through triplicates, each triplicate of size ~6GB, together with three V gene sequence database, each of size 35 MB) took less than three hrs. If a user runs MaxQuant as stand-alone (i.e., not via PASA) and submits to PASA the output peptides list (instead of the raw mass spectrometry data files), the running time was dramatically reduced to no more than a few minutes (using the example V gene sequence database).

## Results

The first step of PASA concerns the analysis of peptides derived from proteomics of serum antibodies. Specifically, PASA provides: (1) A list of peptides and their intensities for each fraction (elution and flow-through), as identified by MaxQuant; (2) The list of peptides that are enriched in the elution relative to the flow-through (by default, over five-fold relative intensity). These peptides correspond to antibodies that are antigen-specific; (3) The mapping of each antigen-specific peptide to the BCR-Seq database. This mapping provides a correspondence between a serum antibody and an antibody clone as found in the BCR-Seq data. Specifically, for some peptides no mapping is found, some map to more than one antibody clone, and some, which we call "informative peptides", are mapped uniquely to a single antibody clone. Informative antigen-specific peptides can be mapped to CDRH3 or to other V gene regions. Peptides that are antigen-specific, informative, and are mapped to CDRH3 are of high interest since they provide a clear link between a secreted antibody to its corresponding antibody clone as found in BCR-Seq data. In the second step of PASA the antibody clones found in the first

step are characterized. Specifically, PASA provides the following information: Isotype distribution (pie chart); CDR3 length distribution (bar plot); V, D, J gene usage (bar plots); VD, VJ, DJ, VDJ combined usage (bar plots); Proteomics intensity with respect to each antibody sequence frequency as observed in the BCR-seq database. For the sake of convenience, the web server includes Gallery and Overview sections, together with a running example.

## Availability and future directions

PASA is freely available for noncommercial users as a web server at https://pasa.tau.ac.il and as a command line application as well. The source code of PASA is written in Python and is available at https://github.com/orenavram/PASA. Joint analysis of Ig-Seq and BCR-Seq provides invaluable insights regarding serum antibody dynamics against a specific antigen. For example, it allows analyzing longitudinal changes in antibody dynamics, studying the affinity maturation of an antibody clone towards a specific antigen, and provides an overall comprehensive map of the humoral immune response following vaccine, disease, or at health. PASA is a user-friendly platform that provides expert and non-expert users with the ability to tune their computations towards their special experimental needs. In the follow-up developments we plan to add advance analysis tools including the ability to analyze experimental data derived from other species and various V gene sequence database.

## Author Contributions

**Conceptualization:** Oren Avram, Aya Kigel, Tal Pupko, Yariv Wine.

**Data curation:** Oren Avram, Aya Kigel, Sharon Kligsberg, Yariv Wine.

**Formal analysis:** Oren Avram, Aya Kigel, Anna Vaisman-Mentesh, Yariv Wine.

**Funding acquisition:** Yariv Wine.

**Investigation:** Oren Avram, Aya Kigel, Anna Vaisman-Mentesh, Shai Rosenstein, Yael Dror, Tal Pupko, Yariv Wine.

**Methodology:** Oren Avram, Sharon Kligsberg, Tal Pupko, Yariv Wine.

**Project administration:** Yariv Wine.

**Resources:** Yariv Wine.

**Software:** Oren Avram, Sharon Kligsberg.

**Supervision:** Tal Pupko, Yariv Wine.

**Validation:** Oren Avram, Aya Kigel, Yariv Wine.

**Visualization:** Oren Avram.

**Writing – original draft:** Oren Avram, Tal Pupko, Yariv Wine.

**Writing – review & editing:** Oren Avram, Tal Pupko, Yariv Wine.

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
