## [Decision Letter · Decision Letter 0]

24 Oct 2020

Dear Dr. Wine,

Thank you very much for submitting your manuscript "PASA: proteomic analysis of serum antibodies web server" for consideration at PLOS Computational Biology.

As with all papers reviewed by the journal, your manuscript was reviewed by members of the editorial board and by several independent reviewers. In light of the reviews (below this email), we would like to invite the resubmission of a significantly-revised version that takes into account the reviewers' comments.

We cannot make any decision about publication until we have seen the revised manuscript and your response to the reviewers' comments. Your revised manuscript is also likely to be sent to reviewers for further evaluation.

Sincerely,

Mihaela Pertea

Software Editor

PLOS Computational Biology

Mihaela Pertea

Software Editor

PLOS Computational Biology

Reviewer's Responses to Questions

**Comments to the Authors:**

Reviewer #1: Summary: In this manuscript, Avram and Kigel and colleagues, report the implementation of the PASA web server that is stated to provide a robust computational platform for the analysis and integration of data obtained from proteomics of serum antibodies. Specifically, PASA enables the mapping of peptides derived from antibodies raised against a specific antigen to corresponding antibody sequences.

Major

Can the authors identify the maxquant version used and the specific maxquant parameters. Furthermore the maxquant parameters files should be provided in the output so that users may reproduce their findings in the future.

Furthermore, the versatility of the webserver remains opaque to the reviewer. Can the webserver also be used if one does not have antigen-specific data? So, just a VDJ-database and MS/MS raw files?

Can the authors provide runtime analyses for common file sizes?

Why is there no command line version which would enable parallelization etc?

What is the novelty of the PASA tool? Have you gained any biological insights using PASA?

Minor

Line 45: please provide original references for the statement that the CDR3H contributes most to antigen binding (for example Xu et al., 2000, Immunity (last author: Mark Davis)).

Line 46: Please provide a reference for the statement: CDRH3 is often used as a unique identifier to determine antibody clonality

https://pasa.tau.ac.il/gallery.html: color scheme for the Isotype Distribution pie chart is not color blind friendly

https://pasa.tau.ac.il/gallery.html: the y-axis “% of total” Percent of total what? All the peptides or ig-related peptides?

https://pasa.tau.ac.il/index.html BCR-Seq DB tooltip: link to the ASAP format goes to the same (index.html) page

https://pasa.tau.ac.il/index.html: “Please notice that since the raw files are (very) heavy.” — English?

Line 92: “Proteomics intensity with respect to each BCR-seq database intensity” — what does BCR-seq database intensity mean?

Reviewer #2: Wine et al. present PASA (proteomic analysis of serum antibodies) a web server for proteomic antibody repertoire analysis. Presumably the server was designed for human samples -no information is provided on whether it might work for antibody repertoires from other species. PASA is very user friendly and enables to process and assign proteomic data using a specimen specific V gene data base. This is a tool that is likely to be very useful for the community and publication is recommended. I wish the V gene DNA sequence was more versatile (only data in ASAP is accepted) but that is a minor complaints.

The manuscript is very well written and to the point.

**Have all data underlying the figures and results presented in the manuscript been provided?**

Reviewer #1: **No: **maxquant parameters file should be provided with the output

Reviewer #2: Yes

PLOS authors have the option to publish the peer review history of their article (what does this mean?). If published, this will include your full peer review and any attached files.

Reviewer #1: No

Reviewer #2: No
---

## [Decision Letter · Decision Letter 1]

6 Dec 2020

Dear Dr. Wine,

We are pleased to inform you that your manuscript 'PASA: Proteomic Analysis of Serum Antibodies web server' has been provisionally accepted for publication in PLOS Computational Biology.

Best regards,

Mihaela Pertea

Software Editor

PLOS Computational Biology

Mihaela Pertea

Software Editor

PLOS Computational Biology

Reviewer's Responses to Questions

**Comments to the Authors:**

Reviewer #1: The authors have addressed all reviewer concerns.

**Have all data underlying the figures and results presented in the manuscript been provided?**

Reviewer #1: Yes

PLOS authors have the option to publish the peer review history of their article (what does this mean?). If published, this will include your full peer review and any attached files.

Reviewer #1: No

---

## [Editor Report · Acceptance letter]

21 Jan 2021

PCOMPBIOL-D-20-01304R1 

PASA: Proteomic Analysis of Serum Antibodies web server

Dear Dr Wine,

I am pleased to inform you that your manuscript has been formally accepted for publication in PLOS Computational Biology. Your manuscript is now with our production department and you will be notified of the publication date in due course.

With kind regards,

Jutka Oroszlan
